# Detecting the Knowledge Domains of Compound Semiconductors

**DOI:** 10.3390/mi13030476

**Published:** 2022-03-20

**Authors:** Qian-Yo Lee, Chiyang James Chou, Ming-Xuan Lee, Yen-Chun Lee

**Affiliations:** 1Department of Biomechatronic Engineering, National Chiayi University, Chiayi 60004, Taiwan; yolee2300@gmail.com; 2Master Program in Entrepreneurial Management, National Yunlin University of Science and Technology, No. 123, University Rd., Section 3, Douliou, Yunlin 64002, Taiwan; 3Graduate Institute of Electronics Engineering, National Taiwan University, Taipei 10617, Taiwan; ken.lee1998@gmail.com; 4Institute of Management of Technology, National Yang Ming Chiao Tung University, Hsinchu City 300, Taiwan; focuslee2@gmail.com

**Keywords:** compound semiconductors, knowledge domain, scientometrics, *CiteSpace*

## Abstract

The development of compound semiconductors (CS) has received extensive attention worldwide. This study aimed to detect and visualize CS knowledge domains for quantifying CS research patterns and emerging trends through a scientometric review based on the literature between 2011 and 2020 by using *CiteSpace*. The combined dataset of 24,622 bibliographic records were collected through topic searches and citation expansion to ensure adequate coverage of the field. While research in “*solar cell*” and “*perovskite tandem*” appears to be the two most distinctive knowledge domains in the CS field, research related to thermoelectric materials has grown at a respectable pace. Most notably, the deep connections between “*thermoelectric material*” and “*III-Sb nanowire* (NW)” research have been demonstrated. A rapid adaptation of black phosphorus (BP) field-effect transistors (FETs) and gallium nitride (GaN) transistors in the CS field is also apparent. Innovative strategies have focused on the opto-electronics with engineered functionalities, the design, synthesis and fabrication of perovskite tandem solar cells, the growing techniques of Sb-based III–V NWs, and the thermal conductivity of boron arsenide (BAs). This study revealed how the development trends and research areas in the CS field advance over time, which greatly help us to realize its knowledge domains.

## 1. Introduction

Semiconductor devices have become an integral part of our lives, where silicon (Si) is the major elemental material that has revolutionized the semiconductor industry for several decades. As Si-based CMOS (complementary metal-oxide-semiconductor) scaling is approaching its physical limits and peak performance, future downscaling of CMOS in accordance with Moore’s law and to meet the demands of the ITRS (International Technology Roadmap for Semiconductors) roadmap will involve new WBG (wide bandgap) materials for the gate dielectrics and the high mobility channels as well as novel structures [1]. Compound semiconductors (CS) are manufactured using WBG materials and they have been playing a crucial role with significant performance advantages in the development of the semiconductor industry [2]. Moreover, CS can be categorized into III–V segment, IV-IV segment, II–VI segment, sapphire and others based on the different requirements of electrical characteristics and a growing range of technology applications [3]. CS has also been applied to the development of photovoltaic fields, because of its good properties as an absorber for solar cells. For example, the Cu(In,Ga)Se_2_ (CIGS) absorber layer was successfully deposited by different substrates, indium tin oxide (ITO), fluorine-doped tin oxide (FTO) and molybdenum (Mo) after optimization of the operating parameters of the deposited film [4].

In recent years, the development of CS has received extensive attention worldwide, because they have been potentially revolutionizing the electrical performance of electronics and changing the landscape of the semiconductor industry. Because CS will attract market attention in the next few decades, it is essential to explore its knowledge domains for scientific frontiers based on the literature. Hence, this paper aimed to detect and visualize the knowledge domains of CS, and to quantify research patterns and development trends in the CS field. We used *CiteSpace* software to investigate the scientific literature collected from the WoSCC (Web of Science Core Collection) [5], and several clusters of research together with significant connections in the CS field in the past decade (2011–2020) which were then reflected.

## 2. Detection and Visualization of Knowledge Domains

A “knowledge domain” can be defined as a distinctive field of research that builds a general base and a sense of advancement of an object by verifying its value and goal to stakeholders [6]. Hence, it can be defined as a broad-based understanding of a particular body of information which allows effective interpretation of data correlations, events or other symptoms in a related process. Visualizing the entire body of scientific knowledge and tracking the recent developments in science and technology have attracted a number of scientists, scholars, researchers, government officials and publishers [7]. The elements ready for knowledge visualization originate from scientific literature. Researchers and professions generally focus on pivotal structural patterns in knowledge discovery, information retrieval and other disciplines that may offer insights into the nature of underlying interrelationships. Furthermore, collaborative networks and intellectual relationships are radical to a knowledge domain [8], and the visual profile of “knowledge networks” helps us to completely understand the intellectual collaborations in a specified knowledge domain [6].

Scientific knowledge may change over time with new data or reevaluation of existing data [9]. Most of these changes are additive, but a few of them are elemental and revolutionary [10]. For instance, several subareas in the CS field have consolidated through the years, ranging from Group III-V elements to Group II-VI elements. Among them, GaAs (gallium arsenide), InP (indium phosphide), InGaAlP (aluminum gallium indium phosphide), etc. have been conventionally used for high-frequency devices and optical devices, while InGaN (indium gallium nitride) has been attracting attention as a material for blue LEDs (light-emitting diodes) and laser diodes, and SiC (silicon carbide) and GaN (gallium nitride) as materials for power semiconductors have been noted and commercialized [11]. Scientific frontiers are not only where one would expect to detect the cutting-edge knowledge and technology but also unsolved mysteries, controversies, battles and debates, and revolutions [10]. Due to the recent progress in bibliographic techniques, scientific indexes and computing power, researchers have been overcoming the difficulties step by step and discovering implicit connections as well as knowledge domains in the literature [2]. The process of exploring the knowledge domains associated with CS is illustrated in the next section.

## 3. Methods

The development of science and technology can be traced by studying the footprints disclosed in scholarly publications. Advances in information visualization offer promising tools for presenting knowledge structures and their development in an increasingly intuitive way [7]. Professor Chen has been developing a tool, named *CiteSpace*, for researchers to easily detect a knowledge domain’s intellectual structures and visualize scientific frontiers [12]. The principle and application of *CiteSpace* were stated as follows.

### 3.1. CiteSpace

*CiteSpace* is a computational tool written in Java for detecting and visualizing research patterns, critical changes and emerging trends in scientific literature [13]. It focuses on the collective behavior of experts and peer scholars in terms of which articles they cite, how often they cited, and contexts in which they cite. Scientific publications are considered relevant if they may orient to a better consensus of the knowledge domain in question. Hence, members of the scientific communities make their contributions that generate a dynamic and self-organizing domain of knowledge. The unit of analysis in *CiteSpace* is a knowledge domain and the design of *CiteSpace* is motivated to achieve two ambitious goals. One is to provide a computational alternative to supplement the traditional systematic reviews and surveys of a body of scientific literature. The other is to provide an analytic tool so that one can study the structure and dynamics of scientific paradigms. *CiteSpace* extracts bibliographic information, especially citation information from the WoS (Web of Science) and generates interactive visualizations [5]. Researchers then can navigate and explore a variety of patterns and trends revealed from scientific publications and develop an integral cognition of the scientific literature. *CiteSpace* conducts the detection and visual profile of a scientific field from bibliographic records in relation to network analysis with different kinds of entities such as co-occurring keywords, co-authors and cited references. Our study focused on the document co-citation analysis with the networks of co-occurring keywords in order to deliver more exhaustive outcome regarding the CS knowledge domains. Each unique node in the knowledge network can be aggregated into clusters on the basis of their strength of interconnection, and each cluster indicates a thematic convergency or a unique specialty. The highly cited landmark articles, the articles with strong citation bursts and the keywords with a strong surge of frequency are also another focal issue for readers. The burst detection is applied to checking whether the cited frequency of an article increases quickly while referring to its peers. For the purposes of our study, we first explored the landmark articles associated with CS. Secondly, we identified the crucial clusters and the emerging topics by means of the citation burst function from the literature for detecting the knowledge domains of CS.

### 3.2. Network Analysis and Visualization

Much of the attention in scientometric analysis has been devoted to document co-citation analysis due to the preferences that citation patterns of references provide, particularly significant insights into the structure and dynamics of scientific paradigms [6]. The input data for *CiteSpace* is a collection of scientific publications related to a specified topic. We then can detect the knowledge domains in a specified topic through its network analysis and visualization. *CiteSpace* can output several types of networks and allow researchers to select Author, Cited References, and Category to form networks of three types of nodes [5]. The default node type is Cited References and the links are co-citation links. In this case, the networks are made of co-cited references. In addition, one of the essential functions of *CiteSpace* is the identification of *Betweenness Centrality* between the pivotal points from the scientific literature [14]. One node’s *Betweenness Centrality* means a measure that denotes the importance degree of the node in a network. *Time-Slicing* function is also a robust tool which provides a temporal outlook to the scientific literature and identifies citation burst. In a word, research fronts mean a collection of articles that are highly cited by other reference publications and usually reveal the properties of one domain’s specificity [15]. Research front terms proposed by *CiteSpace* deliver significant messages of co-citation clusters and provide a universal survey of a knowledge domain with associated networks.

### 3.3. Bibliographic Records

This work gathered the bibliographic source from the WoSCC and determined the timespan of this analysis to the last decade (2011–2020) to ensure the clarity of derived results. The query for CS was set as “*compound semiconductor⁎*” in the topic search. We collected the two datasets of bibliographic records for CS from the WoSCC that includes both SSCI (Social Sciences Citation Index^TM^) and SCIE (Science Citation Index Expanded^TM^) subsidiary databases. The core dataset from the topic-search query resulted in 1413 original research articles. Based on the correlation through citation links, we further acquired the expanded dataset that is a superset of the core dataset with an extra 23,209 bibliographic records. In a nutshell, each article citing at least one original research article from the core dataset can be recognized that it is thematically correspondent to the subject matter from the core dataset [16]. The published items and citations in each year were separately shown in Figure 1. We can see that more than one hundred articles related to the CS field were published in each year and the number of citation increases year by year. Hence, research on CS has been getting more and more attention. We then merged both datasets into one for scientometric review. The total bibliographic sources were thus inputted to *CiteSpace* for the following analysis.

## 4. Results of Scientometric Analysis

The results of scientometric analysis were divided by the following five subsections including document co-citation analysis, identification and interpretation of clusters, most active clusters, references with strong citation bursts, references burst since 2018. Each subsection provided a concise and precise description of the experimental results and interpretation.

### 4.1. Document Co-Citation Analysis

We used *CiteSpace* to detect and visualize the whole dataset of 24,622 bibliographic records that combines both the core dataset and the expanded dataset. We collected the bibliographic records from the WoSCC, and the document co-citation network was then established as listed in Figure 2.

The result based on the document co-citation analysis showed that 557 unique nodes and 1831 links for a one-year time slice existed in this network. These nodes indicate the cited references from the collected articles, and the links within the network indicate the co-citation relationships. Each link color corresponds directly to each time slice. For instance, blue links indicate articles that were co-cited in 2012, and the newest co-citation relationships can be seen as orange or red links. Our study further concluded three key points from Figure 2. At first, the larger node sizes denote that the articles are important ones within the knowledge domains. Secondly, the red rings around a node indicate a citation burst. Finally, the purple rings describe nodes that have a fairly high “*Betweenness Centrality*” in this network.

Table 1 shows the five top-cited articles associated with the term “compound semiconductors” between 2011 and 2020.

The most cited paper was published by Liu, et al. [17] with citation counts of 62, which demonstrated a broadband, high-speed, waveguide-integrated electroabsorption modulator based on monolayer graphene. The second is a review paper by Del Alamo [18], which indicated the limits of traditional silicon CMOS transistor scaling and the outstanding electron transport properties of group III–V compound semiconductors. The third is the work of Li, et al. [19], which fabricated field-effect transistors based on few-layer black phosphorus crystals with thickness down to a few nanometers and demonstrated the potential of black phosphorus thin crystals as a new two-dimensional material for applications in nanoelectronic devices. The fourth is a paper by Liu, et al. [20], which demonstrated the possibility of phosphorene integration by constructing a 2D (two dimensional) CMOS inverter consisting of phosphorene PMOS (p-metal-oxide-semiconductor) and MoS_2_ NMOS (n-metal-oxide-semiconductor) transistors. Finally, Amano, et al. [21] provided a valuable collection of global state-of-the-art GaN research that will inform the next phase of the technology as market driven requirements evolve. Major investments are being made by industrial companies in a wide variety of markets exploring the use of the technology in new circuit topologies, packaging solutions and system architectures that are required to achieve and optimize the system advantages offered by GaN transistors. In conclusion, all five articles indicate CS as a pivotal technology for exploring substantive research issues in global semiconductor industry.

### 4.2. Verification and Justification of Clusters

*CiteSpace* was applied to detecting development trends and research patterns in the body of knowledge as expressed by crucial clusters of articles. Figure 3 shows the crucial clusters of CS research which were labeled with several title terms. It is notable that the size of a cluster’s label is proportional to the size of the cluster.

We can find that there are 95 clusters in the network. To identify the nature of clusters, we extracted noun phrases from the titles of articles that cited the clusters based on one of three selection algorithms–LSI (latent semantic indexing), LLR (log-likelihood ratio) and MI (mutual information). LLR often delivers the most proper result in terms of the uniqueness and coverage of themes correspondent to a cluster. Table 2 shows the top-ranked clusters in order.

As shown in Figure 3, “*solar cell*” and “*perovskite tandem*” are the two largest clusters. “*Perovskite tandem*” is the youngest cluster, and “*thermoelectric material*” and “*hybrid metasurface*” are the two oldest cluster. The silhouette scores of these clusters are more than 0.5, indicating meaningful results. The biggest cluster, “*solar cell*” (cluster #0), consists of 81 members. The three most active citers in this cluster are Shi, et al. [22], Brar, et al. [23] and Liu, et al. [24]. In the light of the titles of these citers in this cluster, research works related to two dimensional (2D) materials form a foundation of the knowledge domain. Researchers who studied 2D opto-electronics especially focused on atomically thin 2D materials in the terahertz domain and hybrid metamaterials with engineered functionalities through the incorporation of graphene, TMDs (transition metal dichalcogenides) and BP (black phosphorus). As expected, this cluster involves a broad range of interests, presenting the interdisciplinary nature of CS. As shown in Figure 3, this cluster has the top-ranked burst item–Novoselov, et al. [25] among all of the clusters, with a burst value of 7.08. Thus, the “*solar cell*” cluster is important to the literature.

The second biggest cluster, “*perovskite*
*tandem*” (cluster #3) in this knowledge domain, has 33 member articles and an average publication year of 2018. This cluster is the newest one in which the three most active citers in this cluster are Li and Zhang [26], Zhang, et al. [27] and Jayawardena*,* et al. [28] accordingly. Because of their remarkable conversion efficiency properties and potentially reduced manufacturing costs, these perovskite tandem solar cells are suitable for developing renewable energy applications [29]. Moreover, these multi-junction (tandem) solar cells (TSCs) consisting of multiple light absorbers with considerably different band gaps show great potential in breaking the Shockley–Queisser (S–Q) efficiency limit of a single junction solar cell by absorbing light in a broader range of wavelengths [26]. Another article worth mentioning is the work of Bouich, et al. [30]. They used antisolvent treatment and optimized thermal annealing to control the nucleation and growth of the MAPbI_3_, and therefore to achieve highly compact perovskite films with large grains, excellent crystalline quality, and very low pinhole density. In recent years, the synthesis, design and fabrication of thin film perovskite tandem solar cells for renewable energy applications has become one of the most popular research subjects within the knowledge domain [31]. Hence, research works related to perovskite tandem solar cells present the most attractive knowledge domain in the CS research field.

The third biggest cluster (cluster #4) is “*III-Sb nanowire*” which has 30 member articles and an average publication year of 2016. The three most active citers within this cluster are Yip, et al. [32], Gazibegovic, et al. [33] and Aseev, et al. [34] respectively. In line with the titles of these citers within this cluster, ultra narrow bandgap III–Sb (antimony) semiconductor nanomaterials provide a unique platform for realizing advanced nanoelectronics, thermoelectrics, infrared photodetection and quantum transport physics. Researchers interested in III-Sb nanowires (NWs) focus particularly on how to control Sb-based III–V NWs growth based on solid-source chemical vapor deposition (CVD), molecular beam epitaxy, metal organic vapor phase epitaxy and metal organic CVD etc. techniques.

The other notable cluster, is the cluster (cluster #6) for “*thermoelectric material*” which consists of 18 member articles and an average publication year of 2015. In the light of the major citing articles [35,36], it can be expected that previous advances push thermoelectric materials to the research forefront of CS development.

Another key cluster appears to the term “*thermal conductivity*”. In most cases, thermal conductivity of BAs (boron arsenide) plays a significant role in the advancement of CS because their ability to conduct and dissipate heat is a critical parameter beyond high electric field strength and electron mobility [37,38]. Thus, thermal conductivity has become a substantial knowledge domain in the CS field.

The term “*2D cesium lead halide*” also appears in a cluster (cluster #10) which has 13 member articles and an average publication year of 2017. Dramatic efforts have been dedicated to the recent progresses in 2D optical structures based on either extrinsic green photoluminescence (PL) from the edges of 2D cesium lead halides or 2D layered halide organic perovskites (LHOPs) [39,40].

An alternative method for detecting the clusters as well as their relationships is to use the “timeline visualization” function as presented in Figure 4.

The most distinct trend in Figure 4 is that most of the documents cited were published after 2016, roughly corresponding to the rise and deployment of CS from a classic type to a great deal of freedom in developing state-of-the-art heterostructures and devices. Interestingly, most of the earliest cited documents in the derived network were published by the journal Applied Physics Letters in 2008 [41,42] and were found in the cluster id #2. Their research works focused on the studies of atomic layer deposition reactions in the CS field. Furthermore, as shown in Figure 3 and Figure 4, the top ranked item by Betweenness Centrality is Li, et al. [43] in Cluster #0, with a centrality score of 0.09. The second one is Liu, et al. [44] in Cluster #0, with a centrality score of 0.08. The third is Cao, et al. [45] in Cluster #3, with centrality of 0.08. These nodes can be regarded as pivotal points, providing significant bridging connections between two research interests.

### 4.3. Most Active Clusters

Figure 4 shows the two clusters, cluster #0 (labeled as “*solar cell*”) and cluster #6 (labeled as “*thermoelectric material*”), with the strongest citation bursts. This indicates that cluster #0 and cluster #3 represent where the primary achievements and contributions of these studies have been since 2016. Table 3 shows the three articles in cluster #3 with the strongest citation bursts.

As shown in Table 3, the highest bursted article in this cluster, Novoselov, et al. [25], reviewed the properties of novel 2D crystals and examined how their properties are used in new heterostructure devices. The second highest bursted article in this cluster, Tan, et al. [46], summarized the unique advances on ultrathin 2D nanomaterials, followed by the description of their composition and crystal structures. The focal subject of the bursted articles to this cluster is the design, synthesis and fabrication strategies of ultrathin 2D nanomaterials for wide ranges of potential applications. Table 4 shows the three articles in cluster #6 with the strongest citation bursts.

As shown in Table 4, the article by Zhao, et al. [48] has the strongest citation burst in this cluster. They proposed a thermoelectric material/technology that converts waste heat into electricity at lower temperatures by introducing small amounts of sodium to the thermoelectric SnSe (tin selenide) in order to enable better conversion efficiencies. The second highest bursted article in this cluster, Jackson, et al. [49], developed the alkali PDT (post deposition treatment) for CIGS (copper indium gallium selenide) solar cells by introducing the heavier alkali elements Rb (rubidium) and Cs (caesium) to increase efficiencies up to 22.6%. The third highest bursted article in this cluster, Tan, et al. [50], reviewed the recent advances in designing high-performance bulk thermoelectric materials for achieving the greatest thermoelectric figure of merit ZT at a constant doping level. A general subject among this group of articles focuses on the design, synthesis and functionalization of thermoelectric materials for maximizing thermoelectric efficiencies.

### 4.4. References with Strongest Citation Bursts

Increasing research interests with regard to the CS knowledge domains are characterized by publications that encountered citation bursts. These articles with citation bursts were based on a total of 24,622 bibliographic records selected from 120,661 associated reference publications. Figure 5 lists the top 10 reference publications with the strongest citation bursts during the period from 2011 to 2020.

As shown in Figure 5, all of the reference publications started to burst in year 2018. The top three references with the strongest citation bursts are Novoselov, et al. [25], Zhao*,* et al. [48], Jackson, et al. [49] and accordingly. As mentioned above, Novoselov, et al. [25] reviewed the properties of novel 2D crystals and investigated how their properties were used in new heterostructure devices. Composed from individual 2D crystals, such devices (e.g., tunneling transistors, resonant tunneling diodes and LEDs) used the properties of novel materials to create functionalities. The burst continued for three years from 2018 to 2020. Zhao, et al. [48] proposed a thermoelectric material/technology that converts waste heat into electricity at lower temperatures by introducing small amounts of sodium to the thermoelectric SnSe in order to enable better conversion efficiencies. Jackson, et al. [49] developed the alkali PDT for CIGS solar cells by introducing the heavier alkali elements Rb and Cs to increase efficiencies up to 22.6%.

## 5. Discussion

We detected the crucial clusters of articles and verified the research patterns and emerging trends from the scientific literature based on the document co-citation analysis and the network visualization by using *CiteSpace*. The top two clusters were labeled as “*solar cell*” and “*perovskite tandem*”, indicating that they are fundamental to the CS knowledge domain. As expected, the two biggest clusters involve a broad range of research interests, presenting the interdisciplinary nature of CS. While research in “*solar cell*” and “*perovskite tandem*” appears to be the two most distinctive knowledge domains in the CS field, research related to thermoelectric materials has grown at a respectable pace. Most notably, the developments of thermoelectrics and NWs in the CS field have demonstrated deep connections with the “*thermoelectric material*” research and “*III-Sb nanowire*” research. The two most active clusters with the strongest citation since 2016 are “*solar cell*” (cluster #0) and “*thermoelectric material*” (cluster #6). This indicates that both the “*solar cell*” cluster and the “*thermoelectric material*” cluster represent where the primary achievements and contributions of these studies have been since 2016. The detected surge of the two keywords– “black phosphorus” and “van der Waals (vdW) heterostructure” in the literature is worth further investigating their contexts and applications in the CS field. The detection showed a rapidly increasing number of works that are especially invested in the analysis of BP materials and vdW heterostructures in CS research. Identically, a detected burst of citations highlights an attractive knowledge domain. For instance, Novoselov, et al. [25] had a research paper that had surged from 2018 to 2020 can contribute to our knowledge base about the complex 2D materials and III-V heterostructures in CS field. In addition, a rapid adaptation of BP field-effect transistors (FETs) and GaN transistors in the CS field is evident.

## 6. Conclusions

Considering the interdisciplinary characteristics of CS, innovative strategies have focused on the opto-electronics with engineered functionalities, the design, synthesis and fabrication of perovskite tandem solar cells, the growing techniques of Sb-based III–V NWs, and the thermal conductivity of boron arsenide (BAs). This study not only delivered an effective way to facilitate the connections between authors and the research themes in the CS community, but also revealed how the development trends and the research areas advance over time, which greatly helped us to realize its knowledge domains. Future works may contribute to a specific sub-field of our study, such as GaN, SiC, and so on.

## Figures and Tables

**Figure 1 micromachines-13-00476-f001:**
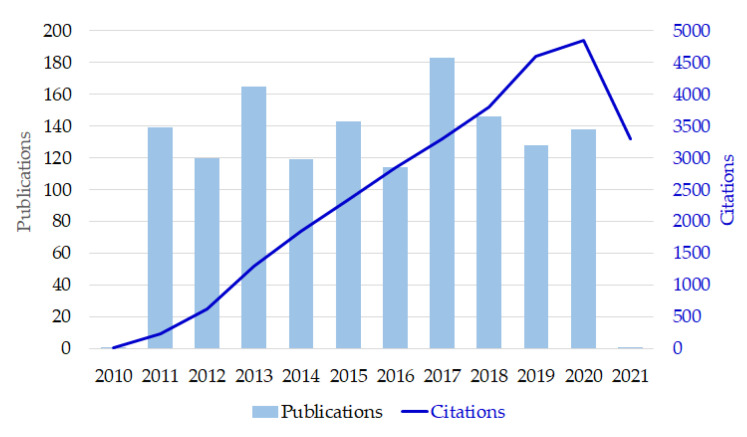
Publications and citations in each year.

**Figure 2 micromachines-13-00476-f002:**
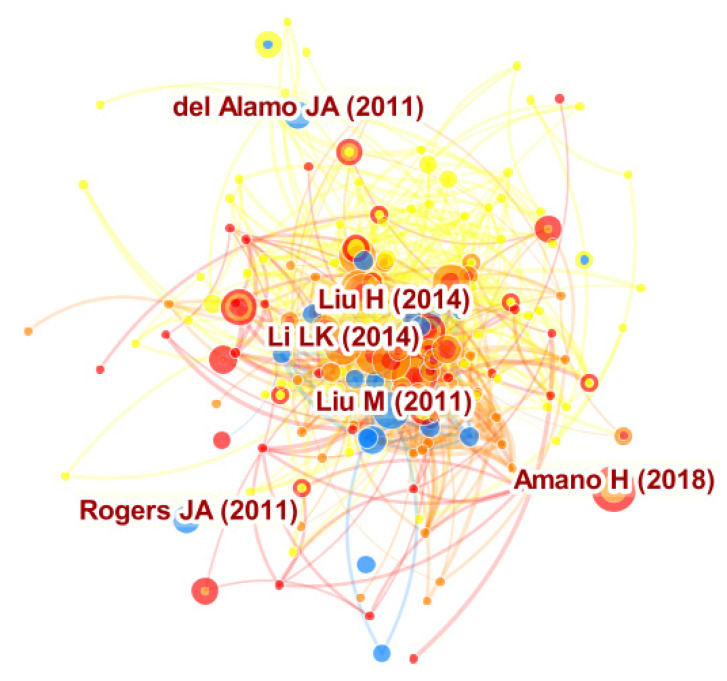
Crucial articles in compound semiconductors (CS).

**Figure 3 micromachines-13-00476-f003:**
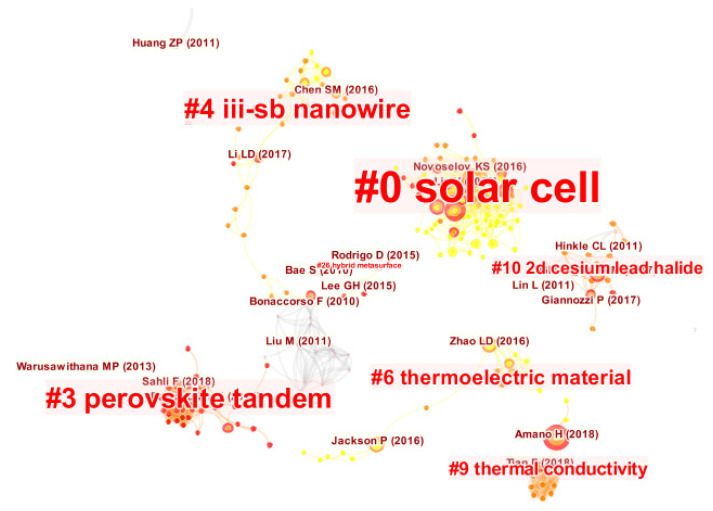
The cluster landscape of CS research by a document co-citation network.

**Figure 4 micromachines-13-00476-f004:**
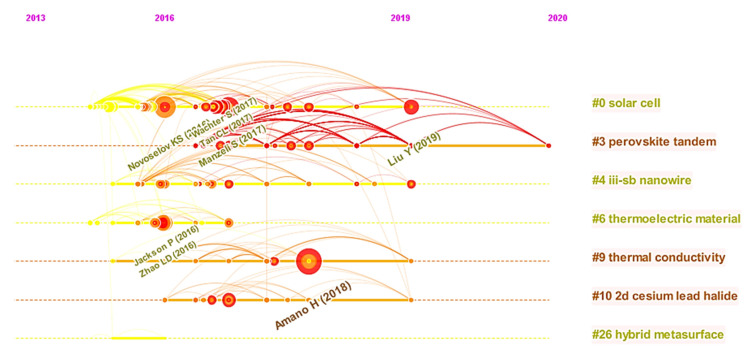
A timeline view of the CS research rendered through a fisheye view.

**Figure 5 micromachines-13-00476-f005:**
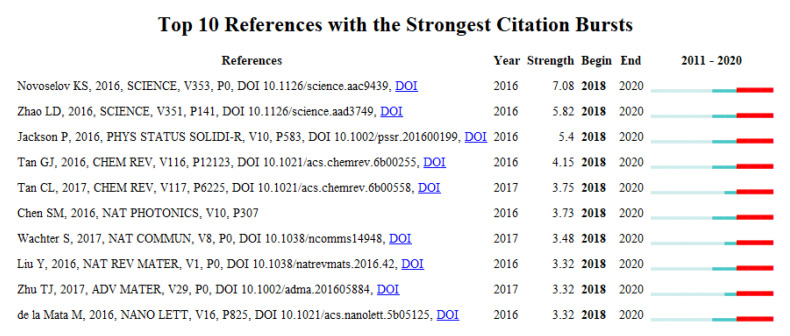
Top 10 reference publications with strong citation bursts.

**Table 1 micromachines-13-00476-t001:** Five critical articles in compound semiconductors (CS).

Cited Frequency	Title	Author	Year	Betweenness Centrality	Journal
62	A graphene-based broadband optical modulator	Liu, M., et al.	2011	0.01	Nature
37	Nanometre-scale electronics with III–V compound semiconductors	del Alamo, J.	2011	0.00	Nature
31	Black phosphorus field-effect transistors	Li, L. K., et al.	2014	0.01	Nat Nanotechnol
29	Phosphorene: An unexplored 2D semiconductor with a high hole mobility	Liu, H., et al.	2014	0.00	ACS Nano
29	The 2018 GaN power electronics roadmap	Amano, H., et al.	2018	0.00	J Phys D Appl Phys

**Table 2 micromachines-13-00476-t002:** Top-ranked clusters in CS.

ID	Size	Silhouette	Label terms (LSI)	Label terms (LLR)	Label terms (MI)	Mean (Cited Year)
0	81	0.929	Black phosphorus, van der Waals (vdW) heterostructure	Solar cell, black phosphorus	Reconfigurable graphene	2016
3	33	0.995	Solar cell, perovskite tandem	Solar cell, perovskite tandem	ZnSnO (zinc-tin-oxide) buffer layer, reconfigurable graphene	2018
4	30	0.971	Hetero-epitaxy, III-Sb (antimony) nanowire	III-Sb nanowire, 2D InSb (indium antimonide) Nanostructure	Si microcone array	2016
6	18	0.993	Thermoelectric material, thermoelectric GeTe (germanium telluride)	Thermoelectric material, thermoelectric GeTe	SnSe (tin selenide) thermoelectric generator	2015
9	14	1.000	Thermal conductivity, HEMT (high electron mobility transistor), BAs (boron arsenide)	Thermal conductivity, BAs	Substrate misorientation, Mg-doped (Magnesium) GaN	2017
10	13	0.870	2D cesium lead halide, extrinsic green photoluminescence, organic spacer substitution	2D cesium lead halide, extrinsic green photoluminescence, organic spacer substitution	Reconfigurable graphene	2017

**Table 3 micromachines-13-00476-t003:** Articles with the strongest citation bursts in cluster #0.

Cited Frequency	Burst	Author	Year	Title	Source
17	7.08	Novoselov, et al. [25]	2016	2D materials and van der Waals heterostructures	Science
14	3.75	Tan, et al. [46]	2017	Recent advances in ultrathin two-dimensional nanomaterials	Chem Rev
13	3.48	Wachter, et al. [47]	2017	A microprocessor based on a two-dimensional semiconductor	Nat Commun

**Table 4 micromachines-13-00476-t004:** Articles with the strongest citation bursts in cluster #6.

Cited Frequency	Burst	Author	Year	Title	Source
14	5.82	Zhao, et al. [48]	2016	Ultrahigh power factor and thermoelectric performance in hole-doped single-crystal SnSe	Science
13	5.40	Jackson, et al. [49]	2016	Effects of heavy alkali elements in Cu(In,Ga)Se_2_ solar cells with efficiencies up to 22.6%	Phys Status Solidi-R
10	4.15	Tan, et al. [50]	2016	Rationally designing high-performance bulk thermoelectric materials	Chem Rev

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
