# Peer review of "Detecting the Knowledge Domains of Compound Semiconductors"

_micromachines, 2022, doi:10.3390/mi13030476_

Round 1
Reviewer 1 Report
Detecting the Knowledge Domain of Compound Semiconduc- 2 tors: A Scientometric Review, The paper and the topic of semiconductors are quite interesting. There are some misses understanding that require some modifications in the paper. After the following corrections paper may be acceptable for publication:
1.The introduction can contains Update references related to the manufacture and the develop of new technologies of semiconductors.
https://doi.org/10.1007/s10854-019-02450-2
2.The figure 1 is not visible graphs. Could the authors improve it and based of this data authors can make conclusion
3.The English and grammatically mistakes should be revise carefully and the monoculture need improvement.
For this part Identification and Interpretation of Clusters, I recommend these references.
Suggested references:
https://doi.org/10.1007/s11837-020-04518-5
5.For Figure 4. A timeline view of the CS research rendered through a fisheye view, can authors check it.
- In this paragraph the authors said : the term “2D cesium lead halide” also represents a cluster (cluster #10) which 260 has 13 member articles and an average publication year of 2017. Dramatic efforts have 261 been dedicated to the recent progresses in 2D optical structures based on either extrinsic 262 green photoluminescence (PL) from the edges of 2D cesium lead halides or 2D layered 263 halide organic perovskites (LHOPs).. what about 2D formadinium and methyl ammonium lead halide that shows good efficiency.
Author Response
1. The introduction can contain Update references related to the manufacture and the development of new technologies of semiconductors. https://doi.org/10.1007/s10854-019-02450-2
Ans.: Thanks for your valuable recommendations. We have strengthened the said parts. Please refer to those paragraphs on a blue background on Line 43-47 and Line 348-350.
2. The figure 1 is not visible graphs. Could the authors improve it and based of this data authors can make conclusion
Ans.: Thanks for your valuable recommendations. We have improved the said parts. Please refer to the revised Figure 1 and those paragraphs on a blue background on Line 141-143.
3. The English and grammatically mistakes should be revise carefully and the monoculture need improvement.
Ans.: Thanks for your valuable recommendations. We have improved the said parts. Please refer to those paragraphs on a gray background throughout the whole paper.
4. For this part Identification and Interpretation of Clusters, I recommend these references. Suggested references: https://doi.org/10.1007/s11837-020-04518-5
Ans.: Thanks for your valuable recommendations. We have added the said parts. Please refer to those paragraphs on a blue background on Line 220-223 and Line 408-409.
5. For Figure 4. A timeline view of the CS research rendered through a fisheye view, can authors check it.
Ans.: Thanks for your valuable recommendations. The “fisheye view” is one of CiteSpace functions developed by Professor Chen (Chen, C. How to use CiteSpace; Leanpub: Victoria, British Columbia, Canada, 2020). The “fisheye view” is provided for the timeline visualization so that researchers can see recent years are displayed with a larger screen estate than earlier years.
6. In this paragraph the authors said: the term “2D cesium lead halide” also represents a cluster (cluster #10) which has 13 member articles and an average publication year of 2017. Dramatic efforts have been dedicated to the recent progresses in 2D optical structures based on either extrinsic green photoluminescence (PL) from the edges of 2D cesium lead halides or 2D layered halide organic perovskites (LHOPs).. what about 2D formadinium and methyl ammonium lead halide that shows good efficiency.
Ans.: Thanks for your valuable recommendations. Because CS begin to gain traction over the next several decades, our study aimed to explore the knowledge domains associated with CS, quantifying research patterns and trends based on the literature from a “macro point of view” by using CiteSpace scientometric tool. Of course, we would briefly summarize the contents of these landmark articles for readers while referring to them in this paper. We did not subjectively make any judgments whether the technology/engineering results of these landmark articles are good or not. We hope you can understand our intention for writing this paper. Many thanks.
Reviewer 2 Report
Although the use of tools such as the one described are of interest to the scientific community, I consider that the paper does not correspond to a review article.
Author Response
1. Although the use of tools such as the one described are of interest to the scientific community, I consider that the paper does not correspond to a review article.
Ans.: Thanks for your valuable recommendations. Instead of traditional review articles, our study used a scientometric software tool – CiteSpace, developed by Processor Chen, to detect and visualize research patterns, critical changes, and emerging trends in scientific literature. It focuses on the collective behavior of experts and peer scholars in terms of which articles they cite, how often they cited, and contexts in which they cite. The design of CiteSpace is motivated to achieve two ambitious goals. One is to provide a computational alternative to supplement the traditional systematic reviews and surveys of a body of scientific literature. The other is to provide an analytic tool so that one can study the structure and dynamics of scientific paradigms. So far, there are a number of papers that reviewed a variety of emerging technologies by using CiteSpace from a macro point of view.
Reviewer 3 Report
The manuscript entitled “Detecting the Knowledge Domain of Compound Semiconductor-2 tors: A Scientometric Review’ is a systematic review of the compound semiconductor research area based on the literature between 2011 to 2020 by using CiteSpace. They investigated the bibliographic cluster visualization and co-citation analysis using cite space in different CS application fields. This manuscript studied and demonstrated the recent trend in the CS application area over time.
This manuscript should be accepted for publication as it is well written, with proper flow and clarity. This work will keep the researcher up-to-date about the current development in this area and their applications and promote others to study their research area based on the citation cluster analysis based on this manuscript using cite space.
Author Response
1. The manuscript entitled “Detecting the Knowledge Domain of Compound Semiconductors: A Scientometric Review’ is a systematic review of the compound semiconductor research area based on the literature between 2011 to 2020 by using CiteSpace. They investigated the bibliographic cluster visualization and co-citation analysis using cite space in different CS application fields. This manuscript studied and demonstrated the recent trend in the CS application area over time.
This manuscript should be accepted for publication as it is well written, with proper flow and clarity. This work will keep the researcher up-to-date about the current development in this area and their applications and promote others to study their research area based on the citation cluster analysis based on this manuscript using cite space.
Ans.: We are very pleased with your positive affirmation. Many thanks.
Round 2
Reviewer 2 Report
I insist on my suggestion, so that it is considered as a regular paper. After this, I give my acceptance to the paper.
Author Response
Thanks for your valuable recommendations. We have revised the article type from “Review” to “Article” on Line 1. The article title was also corrected on Line 2-3.
